# ENTROPY-LENS: UNCOVERING DECISION STRATEGIES IN LLMS

## ABSTRACT

Transformer blocks iteratively refine next-token distributions, yet most interpretability tools analyze hidden states rather than token-space dynamics. We introduce `Entropy-Lens`, a model-agnostic method that tracks the *entropy* of logit-lens predictions across layers, yielding an *entropy profile*: a per-layer, permutation-invariant scalar summary of token prediction dynamic. Entropy differences between consecutive layers act as a proxy for two strategies: *expansion* (more candidates) and *pruning* (fewer candidates). Across model families and scales, entropy profiles show stable family-specific token prediction dynamics and exhibit depth-rescaling invariance. Finally, selectively skipping layers associated with maximal expansion or pruning shows that the two strategies have unequal functional importance for downstream multiple-choice accuracy, with expansion typically being more critical.

## 1 INTRODUCTION

Transformers map a prompt to a distribution over tokens (Vaswani et al., 2023; Shan et al., 2024). Most mechanistic interpretability work studies intermediate representations in embedding space (Skean et al., 2025; Jawahar et al., 2019; Elhage et al., 2022; Skean et al., 2025; Sarfati et al., 2025; Li et al., 2025), while token-space dynamics remain underexplored (Sarfati et al., 2025) due to the vocabulary distribution being high-dimensional and unordered. We tackle this gap by projecting the residual stream into the token space (nostalgebraist, 2020) and analyzing the entropy dynamics of the decoded distributions. This yields a low-dimensional signal that tracks how the set of plausible next tokens grows and shrinks through depth.

**Contributions.** (1) We propose entropy as a sound metric to study the token prediction dynamics and expansion/pruning strategies in the transformer body with single layer and single token level granularity. (2) We show that the particular mixing between these strategies is characteristic of model families, task type and output format. (3) We show that, given a particular model, expansion/pruning strategies have different relative importance for downstream performance, with an overall trend emphasizing expansion.

## 2 ENTROPY-LENS

Analyzing distributions produced by transformers poses two challenges. First, they are high-dimensional, with one probability per vocabulary token. Second, there is no intrinsic notion of order between tokens in the vocabulary. As a result, many standard statistical summaries such as moments or cumulants are unstable or ill-defined when applied to vocabulary distributions.

We address both issues by computing the entropy of each distribution, which has two properties: (i) it is a scalar quantity, and (ii) it is invariant to token permutation. Property (i) addresses the high-dimensionality challenge, and property (ii) makes it suitable for unordered categorical distributions. In the main experiments, we use Shannon entropy $H$, which is defined in this section. Appendix D discusses the broader class of Rényi entropies satisfying the same properties, arguing that Shannon entropy can be taken as a robust parameter-free default.

Let a prompt be tokenized as $S = (t_1, \ldots, t_N)$. Let $x_j^i$ denote the residual-stream activation of token $t_j$ after layer $i$. Following logit-lens, we decode $x_j^i$ through the model head $D$ and apply softmax:

$$y_j^i = W(x_j^i), \quad W := \text{softmax} \circ D. \tag{1}$$

Table 1: Spearman correlation between $\Delta H_i$ and changes in top-$p$ candidate count ($p=0.6$).

| Model | Spearman $\rho$ |
|-------|-----------------|
| Llama-3.2-1B | 0.825 |
| Llama-3.2-3B | 0.737 |
| Gemma-2-2B | 0.880 |
| Gemma-2-9B | 0.854 |

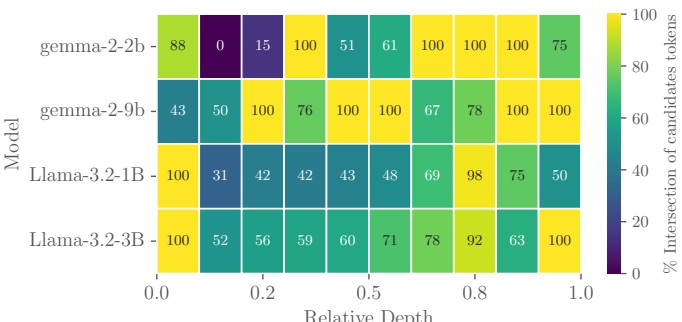

Figure 1: Fraction of shared top-$p$ candidates ($p=0.6$) between consecutive layers. High overlap supports Claim **C2**.

We compute Shannon entropy $H(y) = -\sum_k y_k \log y_k$ and define the *entropy profile* for token $t_j$ as $h_j = \left( H(y_j^1), \ldots, H(y_j^L) \right)$. To summarize a generation of $T$ steps, we aggregate $\{h_1, \ldots, h_T\}$ by concatenation (default) or averaging.

We interpret entropy differences $\Delta H_i := H_i - H_{i-1}$ as: *expansion* if $\Delta H_i > 0$ and *pruning* if $\Delta H_i < 0$, motivated by entropy increasing when probability mass spreads over more candidates.

## 3 VALIDATING THE PROXY

We define "candidates" as the top-$p$ tokens with $p = 0.6$. To justify the interpretation of $\Delta H_i$ as expansion/pruning, we validate two claims:

**C1** $\Delta H_i$ is monotonically related to changes in the number of top-$p$ candidates.
**C2** Consecutive layers share a non-negligible fraction of top-$p$ candidates.

**C1 (monotonicity).** We compute Spearman correlation between $\Delta H_i$ and the change in top-$p$ candidate count across layers. Table 1 shows strong monotonic association.

**C2 (set stability).** Figure 1 shows that consecutive layers retain a large fraction of the top-$p$ candidate set, supporting that entropy changes reflect meaningful pool expansion/pruning rather than arbitrary tail reshuffling.

In summary, the validation of these claims grounds our analysis in a concrete mechanism: *it allows us to interpret increments and reductions of entropy as strategies of expansion and pruning of the candidate set, respectively*. This justifies the use of $\Delta H$ as an interpretable, low-dimensional view of the token prediction dynamics across layers and for individual generations.

## 4 RESULTS

The entropy profiles introduced in the previous section provide a compact view of how intermediate next-token predictions evolve across depth, highlighting phases of expansion and pruning of the candidate set. In this section, we use `Entropy-Lens` to study the token prediction dynamics in LLMs. We show that different families, tasks and output formats are associated with distinct patterns of expansion and pruning of possible next-tokens.

To analyze the differences in entropy profiles, we employ a k-nearest neighbors (kNN) classifier operating directly on the aggregated entropy representations defined in Section 2. The kNN is not used here as a proposed predictive model, but as a diagnostic tool: because a kNN relies entirely on distances between the inputs, its performance provides a direct indication of the distinctness of entropy profiles.

Across all experiments in this section, entropy profiles are extracted from frozen models. Unless otherwise stated, we evaluate classification performance using standard cross-validation protocols and

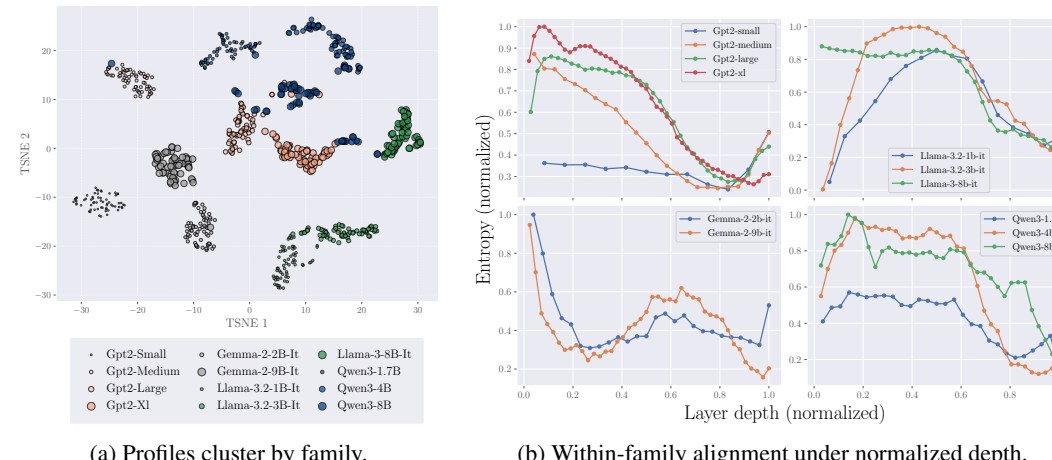

(a) Profiles cluster by family.  (b) Within-family alignment under normalized depth.

Figure 2: Family-specific token prediction dynamics summarized by entropy profiles.

report area under the ROC curve (AUC) scores. Additional implementation details and experiments are provided in Appendix C and B.2, respectively.

In the next subsections, we aim to answer the following research questions (RQs):

**RQ1** Do different models combine expansion and pruning strategies differently in their token prediction dynamics? Do similarities in these dynamics correspond to structural similarities across models?

**RQ2** Do these dynamics change depending on the task?

**RQ3** Do these dynamics change depending on the output format?

**RQ4** Given a particular model, what is the relative importance of one strategy over the other?

**Family-specific dynamics (RQ1).** We first use `Entropy-Lens` to analyze token prediction dynamics across models, assessing whether observed similarities align with structural similarities. We compare entropy profiles across decoder-only LMs spanning multiple families (GPT, Gemma, Llama, Qwen) and scales. Aggregated profiles cluster by *family* rather than parameter count and align when depth is normalized to a relative layer index, indicating *depth-rescaling invariance*. Figure 2 summarizes both effects.

**Task specific dynamics (RQ2).** We then use our framework to assess whether different *task types* have characteristic token prediction dynamics. We consider three task types that differ in the nature of computation they require: *generative* (continue a story), *syntactic* (count the number of words), and *semantic* (extract the main idea/subject).

Across all evaluated models, Table 2a shows that the kNN achieves high AUC, indicating that entropy profiles, and thus token prediction dynamics, present task-relevant structure. Appendix C.3 gives additional details and visualizations, while Appendix B.3 shows the benefits of tracking the entire entropy profile.

Table 2: Results summary. Left: task-type classification results on TinyStories using kNN over aggregated entropy profiles. Right: Format classification results on the topic–format dataset using kNN over aggregated entropy profiles, computed with different Rényi $\alpha$ values. Performance is stable across $\alpha$.

(a) TinyStories

| Model | Size | k-NN AUC (3 classes) |
|---|---|---|
| Gemma-2-it | 2.1B | 97.66 ± 0.47 |
| Gemma-2-it | 8.9B | 98.38 ± 0.50 |
| Llama-3.2-it | 1B | 94.94 ± 0.79 |
| Llama-3.2-it | 3B | 94.77 ± 0.93 |
| Llama-3-it | 8B | 96.10 ± 0.67 |
| Phi-3 | 3.6B | 97.07 ± 0.87 |

(b) Format vs. $\alpha$

| Model | $\alpha$ | k-NN AUC (3 classes) |
|---|---|---|
| | 0.5 | 97.3 ± 1.6 |
| Gemma-2-2B-it | 1.0 | 98.7 ± 1.1 |
| | 5.0 | 98.4 ± 1.7 |
| | 0.5 | 97.8 ± 1.6 |
| Llama-3.2-1B-it | 1.0 | 97.8 ± 2.4 |
| | 5.0 | 96.6 ± 2.6 |

**Output format specific dynamics (RQ3).** Similarly to before, we now use `Entropy-Lens` to assess whether different *formats* of generated text have characteristic token prediction dynamics, independently of topical content. We construct a *topic-format* dataset by prompting models to generate short texts across multiple topics while enforcing one of three different formats among `poem`, `scientific piece` and `chat log`.

Entropy profiles exhibit format-specific characteristics, enabling reliable discrimination between output formats across models (Table 2b). We further find that the results remain stable when replacing Shannon entropy with Rényi entropies across a broad range of $\alpha$ values. To visualize similarities, we project aggregated profiles via PCA. The clear clustering in Figure 9 demonstrates linear separability, confirming that format-specific computation leaves a distinctive signature in the token prediction dynamics.

**Intervening on token prediction dynamics (RQ4).** To test functional relevance, we perform targeted layer skipping in a multiple-choice setting (MMLU) (Hendrycks et al., 2021). For each question, we identify layers with maximal expansion ($\max \Delta H$) or maximal pruning ($\min \Delta H$) and skip the top-$k$ such layers during inference (baseline: skip $k$ random layers). Figure 3 shows that disrupting expansion layers is typically more damaging than disrupting pruning layers; a notable exception appears in Gemma-2-2B-it, where pruning layers are more critical.

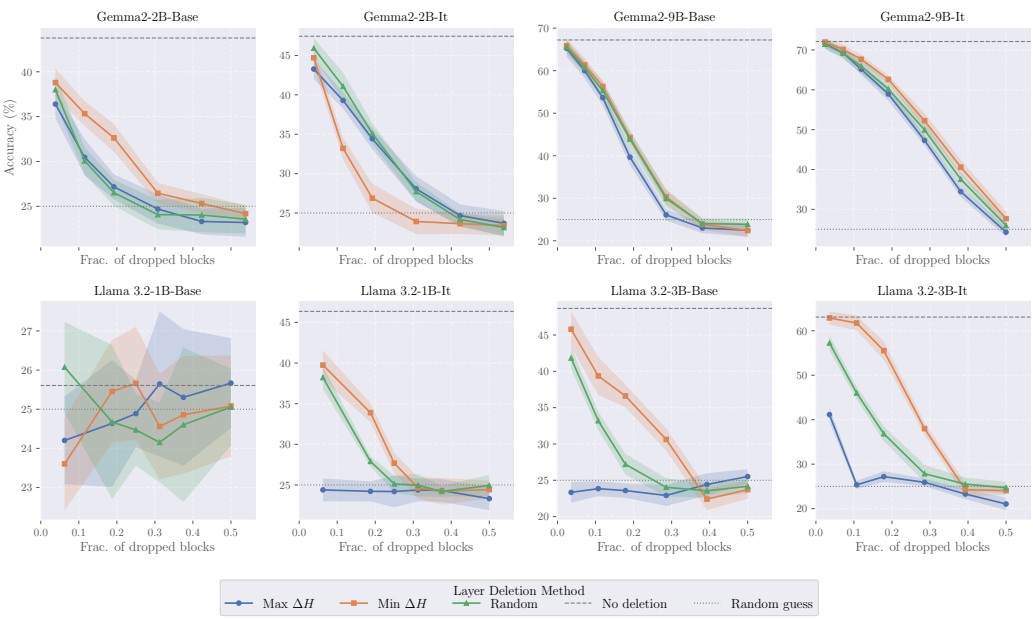

Figure 3: MMLU accuracy under targeted layer skipping. Expansion-disrupting skips ($\max \Delta H$) are usually more harmful than pruning-disrupting skips ($\min \Delta H$).

## 5 DISCUSSION AND LIMITATIONS

Entropy profiles provide a lightweight, per-layer token-space view that reveals stable, family-level decision dynamics and a consistent expansion/pruning decomposition. Open questions include: (i) which architectural/training factors causally determine a family's profile; (ii) how instruction-tuning/RLHF can invert the functional importance of expansion vs. pruning (as observed for Gemma-2-2B-it); and (iii) extensions beyond decoder-only LMs.

## BROADER EXPERIMENTAL EVIDENCE (APPENDIX)

To fit workshop length, we move additional analyses to an appendix, such as Rényi-$\alpha$ robustness and extra implementation details.

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

# ENTROPY-LENS: UNCOVERING DECISION STRATEGIES IN LLMS - APPENDIX

The appendix is organized as follows:

- **Appendix A – Background and Related Work:** We provide background material on information-theoretic quantities and transformer computation, extended methodological details, and an extended discussion of related work.

- **Appendix B – Additional Experiments:** We provide a preliminary exploration of our approach on Vision Transformers (ViTs), showing that entropy profiles can also be extracted and qualitatively interpreted in non-language domains, without any modification to our framework. We also conducted a supplementary experiment on the TinyStories dataset (using the Gemma-2-2b-it model) to determine which transformer blocks are essential for the classification task and whether all blocks are necessary for optimal performance.

- **Appendix C – Evaluation Details:** We provide full details on the datasets, prompt templates, and evaluation protocols used in all experiments. In particular, we highlight details about the model family identification, we specify how prompts were constructed for task classification, output correctness, and format classification tasks. This section also includes hardware information to support reproducibility. We also include all extended experimental results and analyses that do not fit in the main paper, as well as extended discussion items.

- **Appendix D – Theoretical Considerations:** here we detail some of some of the theoretical aspects of the work presented in the main paper. In addition, we outline a heuristic connection between entropy profiles and memorization in transformers. Building on the frameworks of Brown et al. (2021) and Morris et al. (2025), we show how our layer-wise entropy measures can be interpreted as estimates of memorization across depth.

- **Appendix E – Minimal Implementation:** We present a minimal code snippet that reproduces the core entropy profile extraction logic in a few lines of code. While our full codebase offers several optimizations and utilities, this section emphasizes transparency and ease of replication by showcasing the conceptual simplicity of our approach.

## A BACKGROUND AND RELATED WORK

### A.1 BACKGROUND

In this section, we review the fundamental concepts from information theory and key components of the Transformer architecture, with particular emphasis on the residual connections and the information flow across layers. We also briefly categorize the training stages of Large Language Models to clarify the distinction between the base and instruct models used in our study.

#### A.1.1 INFORMATION THEORY

**Shannon entropy.** The main information-theoretic quantity used in our study is *entropy*. Given a discrete[1] random variable $X$ with outcomes $x_i$ and probability mass function $p$, the *Shannon entropy* $H$ of $X$ is defined as $H(X) = -\sum_i p(x_i) \log p(x_i) = \mathbb{E}[-\log p(X)]$. Shannon proved that this function is the only one, up to a scalar multiplication, which satisfies intuitive properties for measuring 'disorder' (Shannon, 1948). These include being maximal for a uniform distribution, minimal for the limit of a Kronecker delta function, and ensuring that $H(A, B) \leq H(A) + H(B)$ for every possible random variable $A$ and $B$. The same function already existed in continuous form in physics, where it linked the probabilistic formalism of statistical mechanics with the more phenomenological framework of thermodynamics, where the term 'entropy' was originally coined (Gibbs, 1902).

**Rényi entropy.** In addition to Shannon entropy, we also consider its generalization known as *Rényi entropy*. Given a discrete random variable $X$ with probability mass function $p$, the Rényi entropy of order $\alpha > 0$, $\alpha \neq 1$, is defined as: $H_\alpha(X) = \frac{1}{1-\alpha} \log \sum_i p(x_i)^\alpha$. This formulation reduces to Shannon entropy in the limit $\alpha \to 1$, and introduces a tunable parameter $\alpha$ that modulates the sensitivity of the entropy to the distribution's tail. Rényi entropy also subsumes many classical

---

[1]Although entropy can be naturally extended to the continuous case with probability *density* functions, we restrict ourselves to the discrete case as it is the most relevant to our study.

descriptors of discrete distributions without intrinsic ordering: with appropriate choices of $\alpha$, it recovers collision entropy ($\alpha = 2$), min-entropy ($\alpha \to \infty$), and max-entropy ($\alpha \to 0$), and it correlates with indices such as the Gini–Simpson index (Rényi, 1961; Jost, 2006). In general, lower values of $\alpha$ give more weight to rare events, while higher values emphasize the most probable outcomes.

**Top-$p$.** Also referred to as Nucleus Sampling (Holtzman et al., 2020), Top-$p$ is a thresholding technique that restricts the distribution to the smallest set of tokens whose cumulative probability mass exceeds a value $p \in (0, 1]$.

### A.1.2 THE TRANSFORMER

**Architecture.** The transformer (Vaswani et al., 2023) is a deep learning architecture widely applied in language modelling with LLMs (Brown et al., 2020) and computer vision (Dosovitskiy et al., 2021). Transformer computations happen through *transformer blocks* and *residual connections*, as exemplified in Figure 4. While various design choices are possible, blocks are usually a composition of layer normalization (Zhang & Sennrich, 2019), attention, and multi layer perceptrons (MLPs). Residual connections, instead, sum the output of layer $i - 1$ to the output of layer $i$.

Inside a single transformer block, the information flows both *horizontally* and *vertically*. The former, enabled by the attention mechanism, allows the token representations to interact with each other. In a language modelling task, for example, this is useful to identify which parts of the input sequence, the sentence prompt, should influence the next token prediction and quantify by how much. The latter vertical information flow allows the representation to evolve and encode different meanings or concepts. Usually, the dimension of the latent space is the same for each block in the transformer. The embedding spaces where these computations take place are generally called the *residual stream*.

**Computation schema.** LLMs are trained to predict the next token in a sentence. That is, given a sentence prompt $S$ with tokens $t_1, \ldots, t_N$, the transformer encodes each token with a linear encoder $E$. Throughout the residual stream, the representation $\mathbf{x}_N$ of the token $t_N$ evolves into the representation of the token $t_{N+1}$, which is then decoded back into token space via a linear decoder $D$, often set to $E^\top$, tying the embedding weights and the decoder. Finally, the logits, the output of $D$, are normalized with $\mathrm{softmax}$ to represent a probability distribution over the vocabulary.

**Instruct models.** Training an LLM typically progresses from self-supervised **pretraining** on vast corpora to **fine-tuning** on structured instructions, often followed by **RLHF** for alignment (Ouyang et al., 2022). While distinction varies, *Chat* tuning focuses on dialogue history, whereas *Instruct* tuning (Wei et al., 2022) targets adherence to specific commands. In this work, we utilize off-the-shelf *Instruct* models (denoted 'it'). We prioritize these over specific chat variants as their flexibility in following command-style prompts aligns better with our experimental setup.

### A.2 EXTENDED METHODOLOGICAL DETAILS

In LLMs, each block operates on the residual stream to map input token sequences to output token distributions. To study their evolution, we introduce `Entropy-Lens`, a **simple**, **scalable**, and **model-agnostic framework** that summarizes layer-wise predictions through their entropy. As shown in Section $C.1$, entropy provides a compact proxy for the number of next-token candidates at each stage of computation.

Conceptually, `Entropy-Lens` can be seen as a dimensionality reduction of transformer computations, compressing complex layer-by-layer predictions into an *entropy profile* that provides a compact representation of how the model progressively expands and prunes its space of plausible continuations. In this section, we detail our methodology (further details in Appendix E.1).

**Entropy profiles.** Formally, let $S = (t_1, \ldots, t_N)$ denote an input sequence of tokens $t_j$ and let $x_j^i$ be the residual-stream activation of token $t_j$ after layer $i$. Similarly to nostalgebraist (2020), decoding this activation through the model's output head and applying a softmax yields an intermediate probability distribution

$$y_j^i = W(x_j^i), \quad W := \mathrm{softmax} \circ D \tag{2}$$

where $D$ is the decoder matrix. We then compute the entropy of this distribution $H(y_j^i)$ to produce a single scalar per layer and token $h_j = (H(y_j^1), \ldots, H(y_j^L))$.

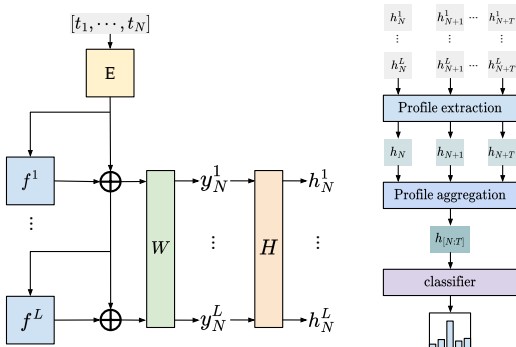

Figure 4: Overview of the `Entropy-Lens` pipeline. Left: intermediate residual-stream activations at each layer are projected into the output space via logit-lens, yielding layer-wise next-token distributions whose entropy is computed. Right: the resulting per-token entropy profiles are aggregated across generated tokens (e.g., by concatenation or averaging) to produce a representation that captures how the model expands and prunes its space of plausible continuations across depth and can optionally be fed to a classifier to assess profile distinctiveness.

Repeating this computation across layers yields a sequence of entropy values which we call *entropy profile*. These profiles compress high-dimensional and unordered token prediction dynamics, providing a low-dimensional, layer-wise information-theoretic representation of the model's computation. Crucially, this construction does not rely on gradients, fine-tuning, or auxiliary probes Tang et al. (2024); Zhao et al. (2025); Zhang et al. (2026), and can be applied to frozen, off-the-shelf models, in a model agnostic way.

**Aggregated entropy profiles.** While the *entropy profile* tracks the candidate pool for a single token, individual predictions are highly variable (Holtzman et al., 2020; Lin et al., 2024). Therefore, model behavior is also analyzed at the level of prompts or generations consisting of multiple tokens. To obtain a stable and informative representation at this level, entropy profiles must be aggregated across tokens.

Specifically, given a prompt, we generate $T$ consecutive tokens autoregressively and extract the profile for each step, yielding a collection of *entropy profiles* $\{h_1, \ldots, h_T\}$. These profiles can be combined using simple aggregation operators to form a single representation of the model's computation for the entire generation.

In our experiments, we primarily use concatenation across tokens, resulting in a matrix-valued representation that *preserves the layer-wise structure of entropy evolution for each generated token*. Alternative aggregation strategies, such as averaging across tokens, are also supported. However, we focus on concatenation as it retains maximal temporal information while remaining straightforward to interpret.

In the following sections, we use `Entropy-Lens` to show that *entropy profiles* make visible systematic patterns in how transformer models expand and contract their candidate pool across depth. These token prediction dynamics exhibit consistent structure within model families (Section C.2), and vary across tasks (Section ??) and output formats (Section C.4). Furthermore, the expansion and pruning phases prove to have different relative importance for downstream task performance (Section C.5).

### A.3 RELATED WORK

**Lenses in LLMs.** Mechanistic interpretability (Bereska & Gavves, 2024) aims to provide a precise description and prediction of transformer-based computations. Common tools in the field are *lenses*, which are a broad class of probes deployed in intermediate steps of the residual stream. For example, logit-lens (nostalgebraist, 2020) uses the model's decoder to project the intermediate activations in the vocabulary space. Tuned-lens (Belrose et al., 2023) refines this technique by training a different affine probe at each layer, instead of only using the pretrained model's decoder. Our framework, `Entropy-Lens`, builds on the Transformer-Lens library (Nanda & Bloom, 2022), which employs

logit-lens to study and characterize LLMs' computations via their decoded version with information theory.

**Information theory in LLMs.** Information theory has been applied to both LLM training and interpretability. For instance, attention entropy collapse is linked to training instabilities (Zhai et al., 2023), and matrix entropy evaluates 'compression' (Wei et al., 2024; de Llano et al., 2025). In a similar vein, Skean et al. (2024; 2025) adapt metrics such as prompt entropy and curvature to evaluate the quality of intermediate representations. Additionally, mutual information helps quantify chain-of-thought effectiveness (Ton et al., 2024). In contrast, `Entropy-Lens` shifts focus to the vocabulary domain, analyzing the entropy evolution of intermediate decoded logits. While Dombrowski & Corlouer (2024) use similar measures to distinguish truthful from deceptive generations under explicit instruction, we uncover broader signatures across model families, tasks, and formats. Regarding memorization, Brown et al. (2021) used Shannon mutual information between data and models, while Morris et al. (2025) extended this using instance-level Kolmogorov theory (see Appendix D.1 for a derivation linking our entropy measure to memorization).

## B  ADDITIONAL EXPERIMENTS

To further test the generality and flexibility of our methodology, we conduct additional experiments beyond the core settings presented in the main text. In particular, we explore how `Entropy-Lens` performs in a different modality: computer vision. Without any architectural adjustment or fine-tuning, we apply the same framework to Vision Transformers (ViTs) and observe that entropy profiles extracted from visual models exhibit qualitatively interpretable structure. These preliminary results suggest that our method may extend beyond language models, but a systematic evaluation across modalities is left for future work.

We further performed a complementary study on the TinyStories dataset, using the Gemma-2-2b-it model, to assess which transformer blocks are most critical for task classification and whether all layers are necessary. In this setting, we compared full entropy profiles with reduced variants obtained from single layers or from equidistant subsets of layers (first, middle, and last). Our results show that the complete entropy profile achieves substantially higher classification accuracy, indicating that information is distributed across depth and cannot be captured by a small subset of layers alone.

### B.1  CORRELATION BETWEEN CHANGES IN ENTROPY AND CANDIDATE COUNTS

Figure 5 demonstrates the relationship between changes in entropy and candidate counts in an example. The reported metrics are calculated by prompting the model with: `"Q: Was Claude Shannon born in an even or odd year?  A:"`

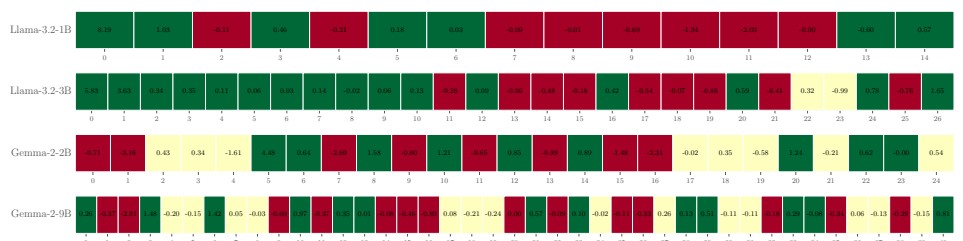

Figure 5: Relationship between changes in entropy and changes in candidate counts. Each box corresponds to a layer, colored in green if the candidate count increases, and red if it decreases. Inside each box, we report the value for $\Delta H$ of the corresponding layer.

### B.2  ENTROPY PROFILES CONTAIN SIGNAL ABOUT ANSWER CORRECTNESS

We finally test whether entropy profiles contain information associated with *answer correctness*. We use the Massive Multitask Language Understanding (MMLU) benchmark (Hendrycks et al., 2021), which consists of multiple-choice questions spanning 57 subjects and varying difficulty levels.

**Setup.** We evaluate two instruction-tuned models (one from the Llama family and one from the Gemma family) on MMLU. To control for prompt framing, we present each question using three prompting styles: **Base**, **Instruct**, and **Humble** (Appendix C.6). For each instance, we extract aggregated entropy profiles from the model generation and label them as correct/incorrect based on whether the selected option matches the ground truth.

**Protocol.** We apply the kNN diagnostic classifier to distinguish correct vs. incorrect profiles and report ROC-AUC under 10-fold cross-validation. To account for class imbalance, we also compare performance against a dummy baseline that samples predictions according to the empirical class proportions.

**Results.** Across prompting styles and both models, entropy profiles yield ROC-AUC values substantially above the baseline (Table 3), indicating that correctness-related signal is present in the entropy dynamics.

Table 3: MMLU correctness for different models. The table reports both the LLM overall accuracy on MMLU and the kNN accuracy in distinguishing correct and wrong answers

| Model | Prompt | LLM-Acc. | kNN AUC |
|---|---|---|---|
| | Base | 50.89 | $73.61 \pm 1.52$ |
| Llama | Humble | 58.51 | $69.90 \pm 1.06$ |
| | Instruct | 60.62 | $67.23 \pm 1.62$ |
| | Base | 56.10 | $71.88 \pm 1.63$ |
| Gemma | Humble | 54.71 | $72.78 \pm 1.15$ |
| | Instruct | 56.38 | $68.36 \pm 1.23$ |

### B.3 ENTROPY PROFILES FROM DIFFERENT BLOCKS

To assess whether entropy profiles from all transformer layers are necessary for model characterization, or if comparable results can be achieved using fewer layers, we conducted an evaluation using different layer subsets. Specifically, we repeated the *TinyStories* experiments (Section **??**) using four different configurations: (1) first layer only, (2) middle layer only, (3) last layer only, and (4) a combination of first, middle, and last layers. We then compared the classification accuracy of k-NN classifiers trained on these reduced entropy profiles against those using complete layer sequences. The results are visible in Table 4.

**Experimental setup.** The k-NN classifier was configured with $k = 11$ neighbors using Euclidean distance as the similarity metric. For sequence generation, we employed a sampling-based approach rather than deterministic decoding which was used for the results reported in the Section **??**.

Table 4: kNN AUC across different sets of considered layers for Gemma-2-2B-it. The results show the benefits of considering the entire entropy profile

| Considered layers | kNN AUC |
|---|---|
| `first-only` | $68.34 \pm 2.68$ |
| `middle-only` | $78.83 \pm 3.07$ |
| `last-only` | $76.78 \pm 2.36$ |
| `first+middle+last` | $86.13 \pm 1.41$ |
| `all` | $90.49 \pm 1.76$ |

### B.4 ENTROPY-LENS FOR VISION TRANSFORMERS

To demonstrate the versatility and robustness of our approach beyond language modeling, we give preliminary results on ViTs and DeiTs.

Using 20 classes from ImageNet-1K (Russakovsky et al., 2015), with 20 images per class, and without any modifications to our framework, we generate the entropy profiles shown in Figure 6. We observe that all profiles start with high entropy values, which then decrease, mostly in the final layers.

This behavior is qualitatively similar to that of GPTs or larger LLaMa models (Section C.2), pointing to a possible common trend across domains as different as image processing and natural language processing.

Focusing on computer vision models, we note that while ViT and DeiT families exhibit qualitatively similar trends, they differ quantitatively—ViTs start with higher entropy values, making them easily distinguishable from DeiTs.

Notably, the only profile that stands out is that of ViT Large (with $\sim 300M$ parameters), compared to the other models analyzed in this section, which have $\leq 86M$ parameters.

For ViT Large, entropy decreases more smoothly, appearing not only as a better approximation of the sharp drop seen in smaller models but possibly following a different behavior entirely, with the entropy decline starting earlier.

We hypothesize a phase transition in entropy behavior as model size increases, occurring somewhere between $87M$ and $307M$ parameters, though a more extensive study would be required to confirm this hypothesis.

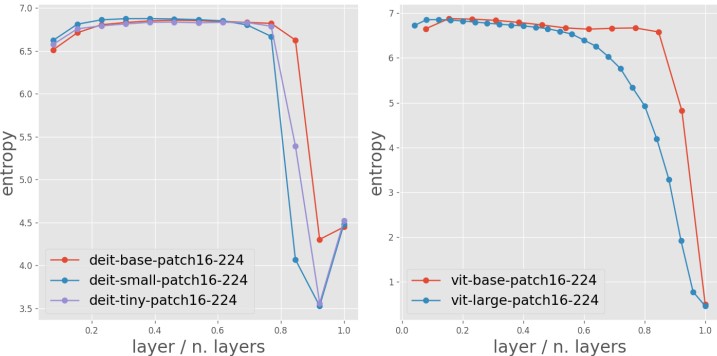

Figure 6: Entropy profiles for ViT model families.

## B.5 STABILITY ACROSS RÉNYI ENTROPY VARIANTS

To qualitatively explore how the Rényi entropy affects the structure of entropy profiles, we compute pairwise cosine similarities between profiles generated with different values of $\alpha$. This analysis is performed on a subset of the *topic-format* dataset (see Appendix C.7), and the resulting similarity matrices are shown in Figure 7.

We observe three distinct regimes as $\alpha$ varies: (1) For very small values of $\alpha$ (e.g., $\alpha < 0.2$), the similarity matrices are nearly flat, with profiles being almost identical across examples. This is expected, as Rényi entropy in this regime weights all tokens with non-zero probability almost equally, and usually all tokens have non-null probability. (2) For large values of $\alpha$ (e.g., $\alpha > 20$), the similarity matrices also flatten. In this case, the entropy becomes increasingly dominated by the few tokens with highest probabilities. Since these sets of top tokens tend to have similar cardinalities (in the limit equal to 1), the profiles collapse into a narrow set of values, losing expressiveness and becoming more sensitive to local fluctuations. (3) Between these extrema lies an informative regime, approximately $0.5 \leq \alpha \leq 20$, where entropy profiles are heterogeneous enough to retain meaningful variation. This is reflected in the standard deviation of the similarity matrices, which peaks in this interval.

We find that the qualitative structure of entropy profiles, shown in Figure 7, and the diagnostic results reported in Section C.7 are stable when replacing Shannon entropy with Rényi entropies for a broad range of $\alpha$ values.

## C EVALUATION DETAILS

This section provides further details on the datasets and prompt templates used to evaluate the effectiveness of entropy profiles in the main experiments. In particular, we describe how we constructed the inputs for three key experimental settings: task type classification using the *TinyStories* dataset, correctness classification using the MMLU benchmark, format classification using the *topic-format* dataset and the intervention strategies experiments.

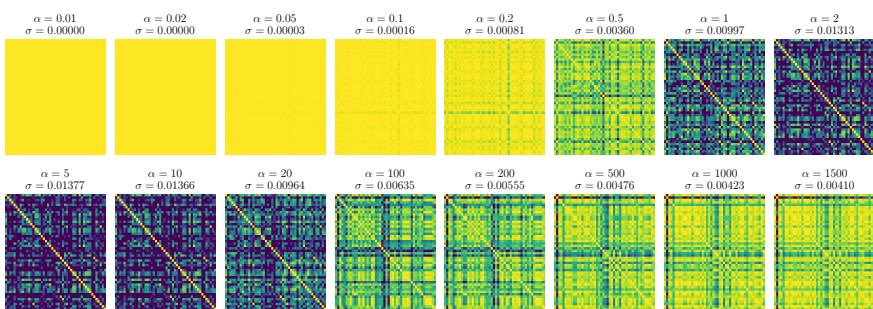

Figure 7: Cosine similarity matrices between entropy profiles computed on a subset of the *topic-format* dataset using different values of $\alpha$ in the Rényi entropy. $\sigma$ denotes the standard deviation of the similarity matrix. Note how similarity flattens for very low and very high $\alpha$, while intermediate values yield more informative profiles.

In all cases, prompt design plays a critical role in ensuring robust comparisons across experimental conditions. To this end, we employed multiple prompt variations. The subsections below report the full set of templates used.

The scripts used to generate these datasets—along with the full codebase to extract entropy profiles and reproduce all experiments—are shared as part of our code release.

Finally, we also provide information about the hardware used to run our experiments to facilitate reproducibility.

## C.1 ENTROPY AND DECISION STRATEGIES

In this section, we motivate the use of entropy from two perspectives. First, Section C.1 establishes it as a robust dimensionality reduction technique for high-dimensional, unordered vocabulary distributions. This addresses the main challenges posed by high-dimensional unordered distributions over token vocabularies. Second, Section C.1 argue that changes in entropy $\Delta H_i$ between a layer $i$ and its predecessor provide a mechanistic interpretation, serving as indicators of expansion and pruning of the next-token candidate set. Finally, Section D generalizes our framework to the broader family of Rényi entropies. In particular, we argue that our findings are robust across a wide and informative range of $\alpha$ values, and not a simple arbitrary choice.

**Entropy as dimensionality reduction.** Analyzing distributions produced by transformers poses two challenges. First, they are high-dimensional, with one probability per vocabulary token. Second, there is no intrinsic notion of order between tokens in the vocabulary. As a result, many standard statistical summaries such as moments or cumulants are unstable or ill-defined when applied to vocabulary distributions.
We address both issues by computing the entropy of each distribution, which has two properties: (i) it is a scalar quantity, and (ii) it is invariant to token permutation. Property (i) addresses the high-dimensionality challenge, and property (ii) makes it suitable for unordered categorical distributions. In the main experiments, we use Shannon entropy $H$, defined in Section **??**. Section D discusses the broader class of Rényi entropies satisfying the same properties, arguing that Shannon entropy can be taken as a robust parameter-free default.

**Entropy changes as a proxy for expansion and pruning strategies.** By definition, entropy is maximized when the probability mass is equally distributed, and minimized when it is concentrated in one event. In our setting, we focus on the entropy of layer-wise next-token predictions extracted via logit-lens (Section 2).

Beyond its absolute value, we also analyze differences in the entropy of intermediate predictions between a layer $i$ and its predecessor $\Delta H_i = H_i - H_{i-1}$. We argue that we can interpret these increments and reductions as strategies of expansions and pruning of the candidate set, respectively. More formally, we define a candidate to be a token appearing in the top-$p$ distribution, with $p = 0.6$. To this end, we need to validate two claims:

**C1** $\Delta H_i$ is monotonically related to changes in the number of possible candidates: positive (negative) entropy variations correspond to increases (decreases) in the number of candidates, with larger variations indicating larger changes.

**C2** Distributions across layers share a non-negligible portion of their candidates.

Claim **C1** ensures that variations in entropy, $\Delta H$, reflect changes in the size of the top-$p$ candidate set, rather than fluctuations confined to the bottom-$(1 - p)$ tail of the distribution. Claim **C2** ensures that the top-$p$ candidate sets remain largely stable across layers. Together, these two claims allow us to interpret increases and decreases of entropy as expansion and pruning of the next-token candidate set.

We validate **C1** by measuring the correlation between $\Delta H_i$ and the change in the number of candidates between layer $i$ and its predecessor. Instead of using Pearson correlation, which assumes a linear relationship, we compute the Spearman rank correlation Gibbons (1993). This allows us to test for a monotonic relationship between entropy variations and changes in the number of top-$p$ candidates, without assuming linearity. Table 1 shows a high correlation, confirming the validity of **C1**. We give further details in Appendix B.1.

To validate **C2**, we measure the percentage of top-$p$ tokens shared by layer $i$ and its predecessor. Figure 1 shows that distributions across layers share many of their candidates, confirming **C2**.

In summary, the validation of these claims grounds our analysis in a concrete mechanism: *it allows us to interpret increments and reductions of entropy as strategies of expansion and pruning of the candidate set, respectively*. This justifies the use of $\Delta H$ as an interpretable, low-dimensional view of the token prediction dynamics across layers and for individual generations.

**Three regimes of Rényi entropy.** In this subsection, we show that our analysis is not tied to an arbitrary entropy definition, but extends to a broad class of dimensionality reductions for categorical distributions.

Through the main body of the paper, we instantiate our framework using Shannon entropy. More generally, our analysis applies to the broader family of Rényi entropies $H_\alpha$, described in Section **??**, which subsumes Shannon entropy as the special case $\alpha \to 1$.

Empirically, we identify an informative regime of $\alpha$ values within which entropy profiles are stable and expressive; importantly, this regime includes the Shannon case $\alpha \to 1$. In Appendix B.5 and Section C.4 (Table 2b), we show that qualitative and quantitative results remain equivalent within this regime. We therefore adopt Shannon entropy as a robust default for the remainder of the paper, avoiding the need to tune additional hyperparameters.

## C.2 MODEL FAMILY CLASSIFICATION

We first use `Entropy-Lens` to analyze token prediction dynamics across models, assessing whether observed similarities align with structural similarities.

**Entropy profiles exhibit family-level patterns.** We compare entropy profiles across 12 decoder-only language models (GPT, Gemma, Llama, Qwen) spanning a wide range of parameter counts (100M to 9B) and depths. Unless otherwise stated, we use a blank prompt and average over multiple generated tokens to obtain a stable profile for each model.

Figure 2a reveals a striking pattern: *entropy profiles cluster by model family rather than by size*. In a low-dimensional projection, models from the same family form coherent groups, while distinct families remain well-separated. This indicates that prediction dynamics are not idiosyncratic to individual checkpoints, but reflect a shared computational structure induced by architectural and training choices.

Qualitatively, different families exhibit distinct entropy regimes, as exemplified in Figure 2b. GPT models start with high entropy and gradually sharpen their predictions, while Llamas display low-entropy initial stages followed by extended high-entropy plateaus and a final refinement. These differences suggest that model families combine expansion and pruning strategies in systematically different ways.

**Depth-rescaling invariance and coarse-to-fine behavior.** A remarkable regularity emerges when comparing models of different scales in the same family. While absolute depth varies substantially,

entropy profiles exhibit a striking alignment when mapped to a *relative layer index* (Figure 2b). This suggests that model scale primarily dictates the *resolution* of the computation rather than its qualitative form. From this perspective, smaller models (less than 4B parameters) appear to implement a coarser discretization of the token prediction dynamics found in their larger counterparts: additional layers serve to refine the signal without fundamentally altering the underlying sequence of expansion and pruning.

Overall, these findings establish entropy profiles as readable fingerprints of model-specific token prediction dynamics. In Section 5.1, we show how entropy profiles can effectively identify both model families and model sizes. Our analysis reveals that entropy profiles exhibit qualitatively distinct patterns across different model families and sizes, as illustrated through scatterplot visualizations. By applying t-SNE dimensionality reduction, we cluster models by family, indicating that entropy profiles capture meaningful structural differences between architectures. To quantitatively assess the classification capabilities of entropy profiles, we employ a k-nearest neighbors classifier ($k = 3$ and euclidean distance) to predict both model families and sizes based on their entropy traces. The classification results are presented in Table 5. To obtain labeled model size categories, we binned models into 4 classes based on parameter count (in billions): <1B, 1-3B, 3-5B, and >5B.

Table 5: F1-scores for model family and model size classification. Each reported value is the mean across 10 runs, with the standard deviation computed over random 50/50 train–test splits.

| Task | Macro F1-score |
| --- | --- |
| model family | 97.99±0.66 |
| model size | 96.31±0.87 |

**Preprocessing Steps.** Since entropy traces vary in length across models due to different layer counts, we apply linear interpolation to standardize all traces to the same length. Additionally, to ensure fair classification performance, we standardize the samples to reduce bias from scaling effects in the entropy profiles, allowing the classifier to focus on the characteristics of each trace.

### C.3    PROMPT TEMPLATES FOR TINYSTORIES TASKS

We now use our framework to assess whether different *task types* have characteristic token prediction dynamics. We consider three task types that differ in the nature of computation they require: *generative* (continue a story), *syntactic* (count the number of words), and *semantic* (extract the main idea/subject).

**Dataset and prompts.** We build prompts using the *TinyStories* dataset (Eldan & Li, 2023). For each task type, we construct prompts by combining a task instruction with a story, using three templates designed to reduce surface-level confounds: *Base* (`task prompt + story`), *Reversed* (`story + task prompt`), and *Scrambled* (either `task prompt + scrambled story` or `scrambled story + task prompt`), where a scrambled story is obtained by randomly permuting the words in a story. The scrambled condition acts as a control to verify that the entropy signature is driven by the task's semantic and syntactic requirements rather than simple lexical patterns. Additionally, to further control for prompt phrasing, we use two semantically equivalent variants for each task instruction, as detailed in Appendix C.3.

**Protocol.** We generate 800 prompts per task type (2400 total), balanced across templates. For each prompt, we extract aggregated entropy profiles as in Section 2 and evaluate task-type separability using a kNN classifier as a diagnostic probe. We report one-vs-rest ROC-AUC under 10-fold cross-validation.

**Results.** Across all evaluated models, Table 2a shows that the kNN achieves high AUC, indicating that entropy profiles, and thus token prediction dynamics, present task-relevant structure. Appendix C.3 gives additional details and visualizations, while Appendix B.3 shows the benefits of tracking the entire entropy profile.

We describe an experimental setup designed to test whether entropy profiles can identify different types of tasks. To this end, we use the *TinyStories* dataset (Eldan & Li, 2023) and construct prompts combining short stories with specific task instructions. Each task type, *generative*, *syntactic*, and

*semantic*, was associated with two distinct natural language formulations, referred to as `task prompts`. These are listed in Table 6. By varying the task prompt, we ensure that our classification results are not simply driven by surface-level textual artifacts, but instead reflect deeper computational signatures captured by the entropy profiles. The same table complements the prompt templates (base, reversed, and scrambled) described in the main text and provides the full set of instructions used to elicit different model behaviors. Figure 8 shows the mean and standard deviation of the entropy profiles for different tasks.

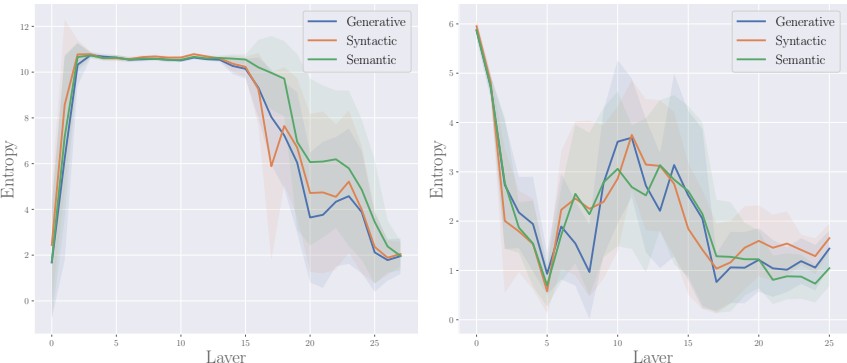

Figure 8: Average entropy profiles with shaded standard deviation for different task types: generative, syntactic, and semantic. These tasks are induced with the prompts described in Appendix C.3. Left: Llama-3.2-it. Right: Gemma-2-it.

Table 6: Prompt templates used for *TinyStories* tasks.

| Task Type | Task prompt |
|---|---|
| Generative | How can the story be continued? 
 Which could be a continuation of the story? |
| Syntactic | How many words are in the story? 
 Count the number of words in the story. |
| Semantic | What is the main idea of the story? 
 Who is the subject of the story? |

## C.4 OUTPUT FORMAT SPECIFIC DYNAMICS (**RQ3**)

Similarly to Section **??**, we now use `Entropy-Lens` to assess whether different *formats* of generated text have characteristic token prediction dynamics, independently of topical content. We construct a *topic-format* dataset by prompting models to generate short texts across multiple topics while enforcing one of three different formats among `poem`, `scientific piece` and `chat log`.

**Dataset construction.** Each prompt is formed by pairing a format instruction with a topic, as described in Appendix C.7. For each generation, we extract aggregated entropy profiles over the generated text according to Section 2.

**Protocol and results.** We evaluate format separability using the same kNN diagnostic setup as above and report ROC-AUC under 10-fold cross-validation. Entropy profiles exhibit format-specific characteristics, enabling reliable discrimination between output formats across models (Table 2b). We further find that the results remain stable when replacing Shannon entropy with Rényi entropies across a broad range of $\alpha$ values. To visualize similarities, we project aggregated profiles via PCA. The clear clustering in Figure 9 demonstrates linear separability, confirming that format-specific computation leaves a distinctive signature in the token prediction dynamics.

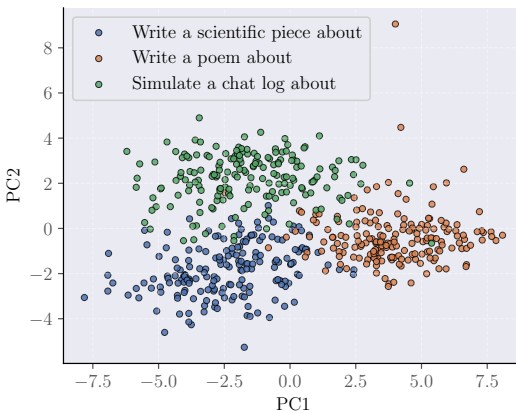

Figure 9: PCA projection of aggregated entropy profiles extracted from the topic–format dataset. Profiles cluster by output format, indicating that different formats are associated with characteristic entropy and token prediction dynamics.

## C.5  Intervening on token prediction dynamics (**RQ4**)

`Entropy-Lens` naturally decomposes token prediction dynamics into two strategies: candidate expansion and pruning (as discussed in Section 2). In this section, we assess the relative importance for downstream accuracy of these two strategies by selectively disrupting layers associated with maximal entropy increases ($\max \Delta H$) and decreases ($\min \Delta H$), respectively. We conduct this analysis in a multiple-choice question setting using the MMLU benchmark (Hendrycks et al., 2021). The choice of this dataset is due to several convenient qualities: its strong fact-based components retain good performance on base models, and it possesses a mild reasoning component, essential in a 1-shot setting. We consider six LLMs, including both base and instruction-tuned (it) variants: Gemma2-2B-base, Gemma2-9B-base, Gemma2-2B-it, Gemma-9B-it, Llama3.2-3B-base and Llama3.2-1B-it.

**Setup.** For each MMLU question, we evaluate the LLM's prediction by the highest probability token among {A, B, C, D} and comparing it to ground truth. To probe the functional roles of expansion and pruning, we identify the top-$k$ layers that maximize or minimize $\Delta H$ per question. These layers are skipped during inference to measure the impact on accuracy. We vary $k$ from 10% to 50% of the total layers in 10% increments. We employ random layer skipping as a baseline to control for the performance degradation caused by model truncation. This allows us to determine if our targeted strategies yield impacts beyond simple architectural degradation. Furthermore, a random baseline avoids the biases inherent in sequential or fixed-order deletion strategies. In this setting, skipping layer $i$ is implemented by setting the contribution of the $i$-th transformer block to the residual stream to zero.

**Results.** Figure **??** shows that expansion ($\max \Delta H$) and pruning strategies ($\min \Delta H$) have different relative impact for downstream performance. Across all Llama variants, we observe a critical dependency: skipping the single layer corresponding to maximal expansion causes performance to collapse to random chance. For Gemma2-2B-Base, Gemma2-9B-Base, and Gemma2-9B-it, we observe a distinct separation between the effects of disrupting expansion versus pruning phases, with the former being more detrimental. In contrast, Gemma2-2B-it exhibits a notable inversion of this dynamic, where pruning phases prove significantly more critical for downstream accuracy than expansion phases. Whether this reversal in functional relevance is a direct byproduct of specific alignment or fine-tuning mechanisms remains an open question for future work.

## C.6  Prompt templates for MMLU correctness classification

In Section B.2, we test whether entropy profiles can distinguish between correct and incorrect answers produced by language models. To construct the dataset for this experiment, we used the Massive Multitask Language Understanding (MMLU) benchmark (Hendrycks et al., 2021), applying three distinct prompt styles to elicit different answer behaviors from the models. Table 7 reports the full prompt templates used in this experiment. Each template presents the same multiple-choice question

in a different instructional format: the **Base** format presents the question directly; the **Instruct** format introduces an explicit instruction to select a single correct answer; and the **Humble** format includes an additional fallback directive encouraging the model to guess randomly if uncertain.

This variation in prompting allows us to control for instruction framing and to evaluate whether entropy profiles can capture response confidence and correctness robustly across different model behaviors. Table 7 complements the description in the main text.

Table 7: Prompt templates used for the MMLU dataset.

| Prompt Type | Prompt |
| --- | --- |
| Base | Subject: {subject} 
 Question: {question} 

 Choices: 
 A. {option_1} 
 B. {option_2} 
 C. {option_3} 
 D. {option_4} 

 Answer: |
| Instruct | The following is a multiple-choice question about {subject}. Reply only with the correct option. 

 Question: {question} 

 Choices: 
 A. {option_1} 
 B. {option_2} 
 C. {option_3} 
 D. {option_4} 

 Answer: |
| Humble | The following is a multiple-choice question about {subject}. Reply only with the correct option. 
 If you are unsure about the answer, reply with a completely random option. 

 Question: {question} 

 Choices: 
 A. {option_1} 
 B. {option_2} 
 C. {option_3} 
 D. {option_4} 

 Answer: |

## C.7   PROMPT CONSTRUCTION FOR THE *Topic-Format* DATASET

To evaluate whether entropy profiles captures stylistic features of generated text, we constructed a custom dataset referred to as the *topic-format* dataset. In this setting, models are prompted to generate short texts on various topics, each constrained to adopt one of three specific formats: `poem`, `scientific piece`, or `chat log`. The goal is to determine whether these formats induce distinct entropy profiles, independently of the topic content.

We generated prompts by pairing 150 distinct topics with the following three format instructions:

- `Write a poem about ...`
- `Write a scientific piece about ...`
- `Simulate a chat log about ...`

Each prompt is constructed by concatenating a format prefix with a randomly selected topic (e.g., `Write a poem about a planet`). The resulting dataset contains 450 prompt completions per model, each paired with its entropy profile computed using Rényi entropy for $\alpha \in \{0.5, 1.0, 5.0\}$.

All generations were performed using a maximum generation length of 256 tokens. We then segmented the output into 8 equal-length windows and computed an entropy profile for each. The resulting data were stored with format labels and used in the classification and visualization tasks discussed in Section C.4 of the main paper.

This setup enables robust testing of the extent to which entropy profiles encode formatting cues, beyond topical content or task semantics.

## C.8   EXPERIMENTAL AND HARDWARE SETUP

The experiments were conducted on:

- A compute node equipped with an NVIDIA L40 GPU, an Intel Xeon Gold CPU, 128 GB of RAM, and running Ubuntu 22.04.
- A compute node equipped with an NVIDIA H100 GPU, an Intel Xeon Platinum CPU, 240 GB of RAM and running Ubuntu 22.04.

The primary software frameworks used were PyTorch, Transformer-Lens, and HuggingFace Transformers. Inference on LLMs was performed using float16 precision for improved efficiency.

## D   THEORETICAL CONSIDERATIONS

**Three regimes of Rényi entropy.** In this subsection, we show that our analysis is not tied to an arbitrary entropy definition, but extends to a broad class of dimensionality reductions for categorical distributions.

Through the main body of the paper, we instantiate our framework using Shannon entropy. More generally, our analysis applies to the broader family of Rényi entropies $H_\alpha$, described in Section **??**, which subsumes Shannon entropy as the special case $\alpha \to 1$.

Empirically, we identify an informative regime of $\alpha$ values within which entropy profiles are stable and expressive; importantly, this regime includes the Shannon case $\alpha \to 1$. In Appendix B.5 and Section C.4 (Table 2b), we show that qualitative and quantitative results remain equivalent within this regime. We therefore adopt Shannon entropy as a robust default for the remainder of the paper, avoiding the need to tune additional hyperparameters.

In addition to the empirical results presented in the main text, we provide here a preliminary theoretical perspective that connects entropy profiles to the literature on memorization in transformers. Our goal is not to give a full formal treatment, but rather to outline how previous definitions of memorization, based on information theory, can be related to the quantities we compute. We first recall the frameworks introduced by Brown et al. (2021) and Morris et al. (2025), and then show how our entropy profiles can be interpreted as layer-wise estimates of memorization.

## D.1 ENTROPY AND MEMORIZATION

**From Shannon to Kolmogorov memorization.** Brown et al. (2021) introduced an information-theoretic framework to quantify memorization in trained models. Given a training data distribution $X$, a family of data-generating processes $\Theta$, and a training algorithm $L : X \mapsto \hat{\Theta}$ mapping training sets to trained models, they define memorization as the mutual information between $X$ and $\hat{\Theta}$

$$\text{mem}(X, \hat{\Theta}) = I(X, \hat{\Theta}) = H(X) - H(X|\hat{\Theta}). \tag{3}$$

This quantity captures how much information about $X$ is retained in the distribution over trained models. It can be decomposed into

$$\text{mem}(X, \hat{\Theta}) = \text{mem}_I(X, \hat{\Theta}, \Theta) + \text{mem}_U(X, \hat{\Theta}, \Theta), \tag{4}$$

where $\text{mem}_I$ measures generalization and $\text{mem}_U$ the unintended memorization (i.e., information about $X$ not attributable to the process $\Theta$).

Building on this formalism, Morris et al. (2025) extended the analysis from distributions to individual instances, moving from Shannon to Kolmogorov information theory. The Kolmogorov complexity of an instance $x$ given model parameters $\hat{\theta}$ is

$$H^k(x|\hat{\theta}) = \min_s \{|s| : f(s, \hat{\theta}) = x\}, \tag{5}$$

where $f$ is a computational model (imagine a decoder) that can take as input $x$ and $\theta$. The exact definition of Kolmogorov complexity is not computable in general. In practice, it can be approximated via arithmetic coding as

$$H^k(x|\hat{\theta}) \approx -\log p(x|\hat{\theta}), \tag{6}$$

where $p(x|\hat{\theta})$ is the predictive probability assigned to $x$ by the trained model. This allowed Morris et al. (2025) to study instance-level memorization, although still focusing on measures computed at the final layer of the model.

**Connecting to entropy profiles.** Our approach provides a complementary perspective. Instead of measuring memorization only at the final layer, we estimate it at every layer when we analyze entropy profiles. To see this connection, recall that in Morris et al. (2025) the term $H^k(x|\hat{\theta})$ is approximated by $-\log p(x|\hat{\theta})$, where $p(x|\hat{\theta})$ is the model's predictive distribution for instance $x$. In our notation, this probability corresponds to a component of the vector $\mathbf{y}_j^i$, the $\text{softmax}$-normalized output obtained for token $t_j$ after block $i$. Averaging this quantity with respect to the distribution $p(x|\hat{\theta})$ yields an estimate of $H(X|\hat{\theta})$. If we further assume that the distribution $\hat{\Theta}$ induced by the training algorithm is sufficiently concentrated around the trained model, this becomes close to $H(X|\hat{\Theta})$, the conditional entropy of the data given the trained model distribution.

Now, rather than computing this value only for the full model, we do so for every intermediate truncation. Let $\hat{\Theta}_i$ denote the sub-model obtained by retaining only the first $i$ layers of the trained transformer and applying an early exit. The corresponding conditional entropies are

$$H(X|\hat{\Theta}_i), \quad i = 1, \ldots, N, \tag{7}$$

and the sequence $\{H(X|\hat{\Theta}_i)\}_{i=1}^N$ constitutes the *entropy profile*. This is equivalent to

$$H(X|\hat{\Theta}_i) = H(X) - I(X, \hat{\Theta}_i), \tag{8}$$

i.e. the negative mutual information between the dataset and the truncated model up to a constant $H(X)$, which is equal for all layers.

This perspective suggests that entropy profiles can be interpreted as measuring how memorization is distributed across depth. Crucially, our empirical results show that this allocation does not follow a simple monotonic trend, as one might have expected a priori. Instead, it varies in a systematic way depending on model family, task, format, and confidence, revealing non-trivial patterns of information storage that had not been documented before.

## E   MINIMAL IMPLEMENTATION

To maximize reproducibility and transparency, we provide a minimal implementation of our framework. While our full codebase includes optimizations and utility functions to streamline analysis across models and datasets, the core idea behind `Entropy-Lens` is conceptually simple and can be expressed in just a few lines of code.

This section presents a compact example that computes the entropy profile of generated tokens using an off-the-shelf language model. Despite its brevity, this snippet captures the essence of our method: extracting intermediate representations, mapping them to vocabulary distributions, and computing their entropies.

### E.1   MINIMAL ENTROPY PROFILE EXTRACTION

The code in Listing 1 demonstrates how to compute an entropy profile for a single prompt using a standard decoder-only transformer. It relies only on model forward passes and the use of `logit-lens`-style decoding. No gradients or fine-tuning are required, but only forward access to intermediate hidden states and the output head.

```python
from transformers import AutoTokenizer, AutoModelForCausalLM
import torch

# Load GPT-2 and set up
tokenizer = AutoTokenizer.from_pretrained('gpt2')
model = AutoModelForCausalLM.from_pretrained('gpt2', device_map="auto").
    eval()
tokenizer.pad_token = tokenizer.eos_token

# Define entropy computation
ln, U = model.transformer.ln_f, model.lm_head
entropy = lambda x: -torch.sum(x * torch.log(x + 1e-15), dim=-1)

# Prepare input
input_text = 'The concept of entropy'
inputs = tokenizer.encode(input_text, return_tensors="pt").to(model.
    device)

# Generate with hidden states
outputs = model.generate(inputs,
    do_sample=True,
    max_new_tokens=32,
    output_hidden_states=True,
    return_dict_in_generate=True,
    pad_token_id=tokenizer.pad_token_id
)

# Stack hidden activations and compute entropy signature
activations = torch.vstack([
    torch.vstack(h).permute(1, 0, 2) for h in outputs.hidden_states
])
entropy_signature = entropy(U(ln(activations)).softmax(dim=-1))
```

Listing 1: A minimal Python implementation of Entropy-Lens using Huggingface models and Pytorch.

