# OpenReview forum: "Entropy-Lens: Uncovering Decision Strategies in LLMs"
_ICLR.cc/2026/Workshop/Sci4DL — Sci4DL 2026_

### Official Review · Reviewer_DzeZ · 2026-02-27

**Fit:** 3
**Significance:** 3
**Confidence:** 2

**Summary:**

This paper introduces Entropy-Lens, a model agnostic tool that tracks the entropy of logit-lens predictions across layers and produces per layer and per token entropy profiles. The layer to layer entropy differences are interpreted as expansion (more candidates, entropy increases) and pruning (fewer candidates, entropy decreases) strategies. The authors validate this proxy by showing strong correlation between entropy changes and top-p candidate count changes and high overlap between consecutive layers' candidate sets. They use kNN classifiers on the aggregated entropy profiles and they demonstrate that:
1) Profiles cluster by model family rather than size (which exhibits depth-rescaling invariance)
2) Different task types produce distinct profiles
3) Output formats (poem, scientific, chat, etc) have distinct entropy signatures
4) Selectively skipping expansion layers is more damaging to MMLU accuracy than skipping pruning layers

**Strengths:**

- The core idea of tracking entropy of decoded intermediate reps across depth is simple, model agnostic, requires no training and provides surprisingly rich insights. The method is easy to implement and broadly applicable
- The authors carefully validate their proxy interpretation with two concrete claims (C1: monotonicity, and C2: set stability) which is backed by Spearman correlations and candidate overlap analysis.
- The paper covers a wide range of experiments (model families, classification, task type, format, layer skipping, etc) which provides a comprehensive view of the tool's utility
- The finding that entropy profiles align across model scales within a family when normalised ot relative depth is an interesting empirical regularity that may suggest something fundamental about how the architecture processes information
- The layer skipping experiment on MMLU goes beyond descriptive analysis to show that expansion/pruning layers have different functional importance and provides some downstream consequences

**Suggestions:**

- The paper does not clearly report how many prompts underlie each experiment or whether family/task/format clustering holds across diverse prompts. The observed differences could be artifacts of specific prompts rather than real family differences. The paper should show clustering holds at larger prompt scale and when controlling for generated content
- Logit-lens is known to be noisy in early layers but the method treats entropy values as equally meaningful at all layers. The C1/C2 validation shows entropy tracks candidate counts but doesn't address whether early layer values are interpretable or just noise
- The kNN classifier is the primary quantitative evidence for family/task/format claims, but only AUC is reported. High AUC can result from tiny but consistent differences that have no functional significance. Effect sizes are not reported for the profile differences, making it hard to assess whether the observed clustering reflects something meaningful or just minor statistical separability. Confusion matrices and analysis of misclassified cases would help
- Regarding layer skipping: when multiple layers are skipped simultaneously, the damage may reflect dependencies across layers rather than the independent importance of each skipped layer. Additionally, zeroing a layer's output creates out-of-distribution inputs for all subsequent layers, so the observed accuracy drop could partly reflect architectural disruption rather than the functional importance of expansion vs pruning specifically

---

### Official Review · Reviewer_Ay84 · 2026-02-27

**Fit:** 2
**Significance:** 2
**Confidence:** 3

**Summary:**

This paper builds directly on Logit Lens. Logit Lens already allows us to inspect the vocabulary probabilities predicted at each layer and observe how sharp or spread out they are. The difficulty, however, is that those distributions are extremely high dimensional. We are dealing with raw probabilities over the entire vocabulary, which makes it hard to visualize, compare, or extract meaningful patterns across layers, models, or tasks.

Entropy-Lens proposes a scalar compression of those distributions. Instead of analyzing raw logits or full probability vectors, it constructs an entropy profile. This is a per-layer scalar summary that is permutation invariant and reflects how sharp or flat the token probability distribution is at each point in the network. The move is essentially to reduce the distribution to a one-dimensional trajectory across depth.

To me, the main contributions are:
	1.	Proposing entropy as a clean scalar metric for studying logit-lens distributions.
	2.	Formulating hypotheses about how these entropy profiles vary across model families, task types, and output formats.
	3.	Using this per-layer metric to investigate which layers matter for downstream performance by selectively skipping those associated with strong expansion or pruning.

I think the idea is directionally very good. There is clearly enough material here for a workshop paper. But I have some substantial concerns about framing and justification.

**Strengths:**

1. Clean Conceptual Move

The scalarization of logit-lens distributions into entropy profiles is conceptually neat. It provides a simple and interpretable summary of something that is otherwise unwieldy. The entropy trajectory across layers is easy to compare across models and tasks, and it allows the authors to cluster models and construct hypotheses about “expansion” and “pruning.”

2. Proxy Validation

The section validating the expansion and pruning interpretation is well structured. They show that changes in entropy correlate with changes in top-p candidate counts and that consecutive layers share candidates. While this result feels intuitive, I appreciate that they formalized and empirically verified it instead of assuming it.

3. Breadth of Empirical Analysis

The paper examines multiple axes: model family, task type, output format, and layer skipping. The scope is broad, and I do think the direction of exploring internal dynamics through simple metrics is valuable.

**Suggestions:**

1. Why Entropy Specifically?

My biggest issue is that the paper does not sufficiently justify why Shannon entropy is the right compression.

The motivation given is that vocabulary distributions are high dimensional and unordered, and that many classical statistics are unstable. That is fair. But there are already existing scalar reductions for logit-lens style analyses:
	•	Cross-layer KL divergence.
	•	Prediction difference metrics.
	•	Divergence-to-final-layer approaches.
	•	Tuned Lens style layerwise translations.
	•	Other cross-layer statistics.

KL divergence between layers is also a scalar and also well defined for unordered categorical distributions. So I do not see a clear argument for why entropy is superior or even preferable to those existing compressions.

The paper treats entropy as a natural choice, but it does not compare it against alternative reductions or explain why this metric captures something fundamentally different. This leaves me unconvinced that entropy is the key insight rather than simply one possible scalar summary among many.

I am not arguing that entropy is incorrect. I am saying that the comparative justification is missing.

⸻

2. Proxy Validation Feels Expected

While Section 3 is methodologically sound, the validation that entropy increases correspond to broader candidate sets and decreases correspond to narrower ones feels almost tautological. It functions more as a sanity check than as a surprising empirical finding.

I appreciate that they did it. But it does not strongly strengthen the conceptual claim.

⸻

3. Appendix Contains Core Material

It feels clear that this was a longer paper adapted to workshop length. That is fine. However, key methodological explanations, such as aggregation of entropy profiles and detailed definitions, are pushed to the appendix.

Those explanations are central to understanding the method and should have been summarized more clearly in the main paper.

⸻

4. Research Questions Lack Framing Depth

The research questions are interesting but not sharply framed.

For example:

RQ2 asks whether the “dynamics” change depending on the task. The word dynamics is doing too much work. It should be explicitly stated that we are referring to entropy trajectories of decoded token distributions.

RQ3 asks whether dynamics change depending on output format. The experiments show separability between poems, scientific text, and chat logs. But what does that mean for visible behavior?

Do entropy differences correlate with quality?
Do they explain stylistic properties?
Do they reflect precision, ambiguity, creativity?

Right now, the paper demonstrates separability but does not connect that separability to interpretable output properties.

⸻

5. Expansion Versus Pruning Framing

The question about relative importance of expansion versus pruning is interesting, but I am not convinced by its framing.

Why is the key comparison one strategy over the other?
Why not ask how different patterns of entropy evolution relate to different observable behaviors?

The layer skipping experiment shows that removing expansion-heavy layers hurts performance more in some cases. But I am not fully convinced that this establishes a meaningful conceptual distinction rather than a layer importance artifact.

⸻

Suggestions

1. Stronger Comparative Justification

The paper would benefit from explicitly comparing entropy against alternative scalar reductions such as KL divergence. Even a small empirical comparison would strengthen the claim that entropy is not just convenient, but informative in a distinct way.

2. Clarify the End Goal

The framing should make clearer why entropy trajectories matter beyond being clusterable.

If entropy profiles differ across formats, what insight does that provide?
If expansion layers matter more, what does that imply about reasoning or uncertainty?

The current presentation is descriptive. It would benefit from a stronger interpretative or interventional framing.

3. Make Research Questions Self-Contained

Each research question should define what is meant by dynamics. They should be self-readable without relying on internal jargon.

⸻

Overall Impression

I think this paper is directionally strong and in the right space. I enjoy empirical analyses of internal dynamics, and I like the idea of simplifying logit-lens signals into something tractable.

However, I am not fully convinced that entropy is uniquely justified as the central metric, nor am I convinced that the current framing clearly articulates why the questions being asked are the most interesting ones.

With sharper framing and stronger justification of the metric choice, this could be a much stronger contribution.

---

### Meta-Review · Area_Chair_dNNs · 2026-03-02

**Recommendation:** Accept

**Metareview:**

This paper proposes a method that tracks the entropy of logit-lens predictions across layers, and studies it across model families, task types, and output formats. The methodology is sound and the findings can be of interest for the workshop.

---

### Decision · Program_Chairs · 2026-03-02

Accept